# Control of brown adipose tissue adaptation to nutrient stress by the activin receptor ALK7

Patricia Marmol[1], Favio Krapacher[1], Carlos F Ibáñez[1,2,3,4]*

[1]Department of Neuroscience, Karolinska Institute, Stockholm, Sweden;
[2]Department of Physiology, National University of Singapore, Singapore, Singapore;
[3]Life Sciences Institute, National University of Singapore, Singapore, Singapore;
[4]Stellenbosch Institute for Advanced Study, Wallenberg Research Centre at Stellenbosch University, Stellenbosch, South Africa

**Abstract** Adaptation to nutrient availability is crucial for survival. Upon nutritional stress, such as during prolonged fasting or cold exposure, organisms need to balance the feeding of tissues and the maintenance of body temperature. The mechanisms that regulate the adaptation of brown adipose tissue (BAT), a key organ for non-shivering thermogenesis, to variations in nutritional state are not known. Here we report that specific deletion of the activin receptor ALK7 in BAT resulted in fasting-induced hypothermia due to exaggerated catabolic activity in brown adipocytes. After overnight fasting, BAT lacking ALK7 showed increased expression of genes responsive to nutrient stress, including the upstream regulator KLF15, aminoacid catabolizing enzymes, notably proline dehydrogenase (POX), and adipose triglyceride lipase (ATGL), as well as markedly reduced lipid droplet size. In agreement with this, ligand stimulation of ALK7 suppressed POX and KLF15 expression in both mouse and human brown adipocytes. Treatment of mutant mice with the glucocorticoid receptor antagonist RU486 restored KLF15 and POX expression levels in mutant BAT, suggesting that loss of BAT ALK7 results in excessive activation of glucocorticoid signaling upon fasting. These results reveal a novel signaling pathway downstream of ALK7 which regulates the adaptation of BAT to nutrient availability by limiting nutrient stress-induced overactivation of catabolic responses in brown adipocytes.

*For correspondence:
carlos.ibanez@ki.se

Competing interests: The authors declare that no competing interests exist.

## Introduction

The adipose depots in mammals consist mainly of white (WAT), beige (also known as brite) and brown (BAT) adipose tissues (*Frontini and Cinti, 2010*; *Petrovic et al., 2010*; *Wu et al., 2012*; *Rosen and Spiegelman, 2014*). WAT stores energy in the form of triglycerides, which can be mobilized in time of higher energy expenditure or nutrient scarcity. Fat accumulation in WAT is an anabolic process, mainly regulated by insulin, while fat breakdown by lipolysis can be considered as catabolic and is predominantly controlled by catecholamines. The mitochondria of BAT contain Uncoupled Protein 1 (UCP1) and defends body temperature against low environmental temperature producing heat through non-shivering thermogenesis (*Cannon and Nedergaard, 2004*). Under certain conditions, subcutaneous WAT depots can also develop UCP1-expressing cells, referred to as beige or brite adipocytes (*Wu et al., 2013*; *Nedergaard and Cannon, 2014*). Cold sensing in mammals results in the transduction of a signal from the central nervous system to the sympathetic nerve endings in BAT. Adrenergic stimulation in BAT triggers the release of long-chain fatty acids from cytoplasmic lipid droplets (*Cannon and Nedergaard, 2004*), which act as both the main energy substrate for thermogenesis as well as activators of $H^+$ transport activity in UCP1 (*Fedorenko et al., 2012*). Dissipation of the mitochondrial $H^+$ gradient generated during substrate oxidation results in

heat production at the expense of ATP synthesis (*Nicholls and Locke, 1984*). Other origins of fatty acids for thermogenesis include WAT lipolysis as well as dietary fat. During non-shivering thermogenesis, cytosolic glucose oxidation is also greatly increased to counterbalance the decrease in ATP synthesis that results from uncoupling (*Hao et al., 2015*). While mainly driven by the central nervous system, non-shivering thermogenesis is also known to be regulated by secreted factors (*Villarroya and Vidal-Puig, 2013*).

Due to the high amounts of energy consumed by BAT, non-shivering thermogenesis is highly sensitive to nutrient availability and functions optimally during well-fed conditions. In periods of nutrient scarcity, mammals are able to reduce thermogenesis as an adaptive, energy-saving mechanism by entering torpor, a state characterized by decreased activity and body temperatures lower than 32°C (*Geiser et al., 2014*). Mice can enter torpor when confronted with severe nutrient stress, such as during prolonged fasting, and can be exacerbated by cold exposure (*Oelkrug et al., 2011*). During a situation of nutrient stress combined with low ambient temperature, animals must reconcile the requirements of high energy demanding organs, such as the brain, with the need to maintain body temperature. However, our understanding of the mechanisms that control this balance is limited. Mechanisms underlying responses to nutritional stress are better understood in liver and muscle, and include the induction by glucocorticoids of Kruppel Like Factor 15 (KLF15), a key regulator of nutritional adaptations during fasting, such as liver gluconeogenesis and amino acid catabolism in muscle (*Yamamoto et al., 2004*; *Gray et al., 2007*; *Shimizu et al., 2011*). Whether BAT employs similar or different mechanisms to adapt its thermogenic activity to fluctuations in nutrient availability is currently unclear (*Himms-Hagen, 1995*).

ALK7, encoded by the *Acvr1c* gene, is a type I receptor of the TGF-β receptor superfamily that mediates the activities of a diverse group of ligands, including activin B, growth and differentiation factor 3 (GDF-3) and Nodal (*Rydén et al., 1996*; *Reissmann et al., 2001*; *Andersson et al., 2008*). In rodents as well as humans, ALK7 expression is enriched in tissues that are important for the regulation of energy homeostasis, including adipose tissue (*Andersson et al., 2008*), pancreatic islets (*Bertolino et al., 2008*), endocrine gut cells (*Jörnvall et al., 2004*) and the arcuate nucleus of the hypothalamus (*Sandoval-Guzmán et al., 2012*). In white adipose tissue (WAT), previous studies have shown that ALK7 signaling facilitates fat accumulation under conditions of nutrient overload, by repressing the expression of adrenergic receptors, thereby reducing catecholamine sensitivity (*Guo et al., 2014*). Accordingly, mutant mice lacking ALK7 globally, or only in adipocytes, are resistant to diet-induced obesity (*Andersson et al., 2008*; *Yogosawa et al., 2013*; *Guo et al., 2014*). Recent studies have identified polymorphic variants in the human *Acvr1c* gene which affect body fat distribution and protect from type II diabetes (*Emdin et al., 2019*; *CHD Exome+ Consortium et al., 2019*), indicating that ALK7 has very similar functions in humans as in rodents. Whether ALK7 is required for normal BAT function is currently unknown.

In the present study, we have used BAT-specific mouse mutants lacking ALK7 in brown adipocytes to elucidate in vivo roles of ALK7 in BAT physiology. In the process, we uncovered a novel signaling pathway involving glucocorticoid signaling, KL15 and POX, which contributes to regulate the adaptation of BAT physiology to variations in nutritional status.

## Results

### Fasting induces abnormally increased fat catabolism in BAT of *Ucp1*[CRE]:*Alk7*[fx/fx] mutant mice lacking ALK7 in brown adipocytes

Expression of *Acvr1c* mRNA (encoding ALK7) was detected in interscapular BAT (iBAT) of young adult male mice at levels comparable to those found in inguinal WAT (iWAT), although lower than *Acvr1c* mRNA expression in epididymal WAT (eWAT) (*Figure 1A*). No *Acvr1c* mRNA expression could be detected in liver. The level of *Acvr1c* mRNA was low in cells isolated from BAT stromal vascular fraction (SVF), containing precursors of brown adipocytes, but was markedly upregulated after in vitro differentiation into brown adipocytes, reaching levels comparable to those found in mature adipocytes freshly isolated from BAT (*Figure 1B*). In order to investigate cell-autonomous functions of ALK7 in BAT, we generated mice lacking this receptor specifically in brown adipocytes by crossing *Alk7*[fx/fx] mice (*Guo et al., 2014*) with *Ucp1*[CRE] mice (*Kong et al., 2014*), expressing CRE recombinase under regulatory sequences of the gene encoding Uncoupling Protein 1 (*Ucp1*). In the resulting

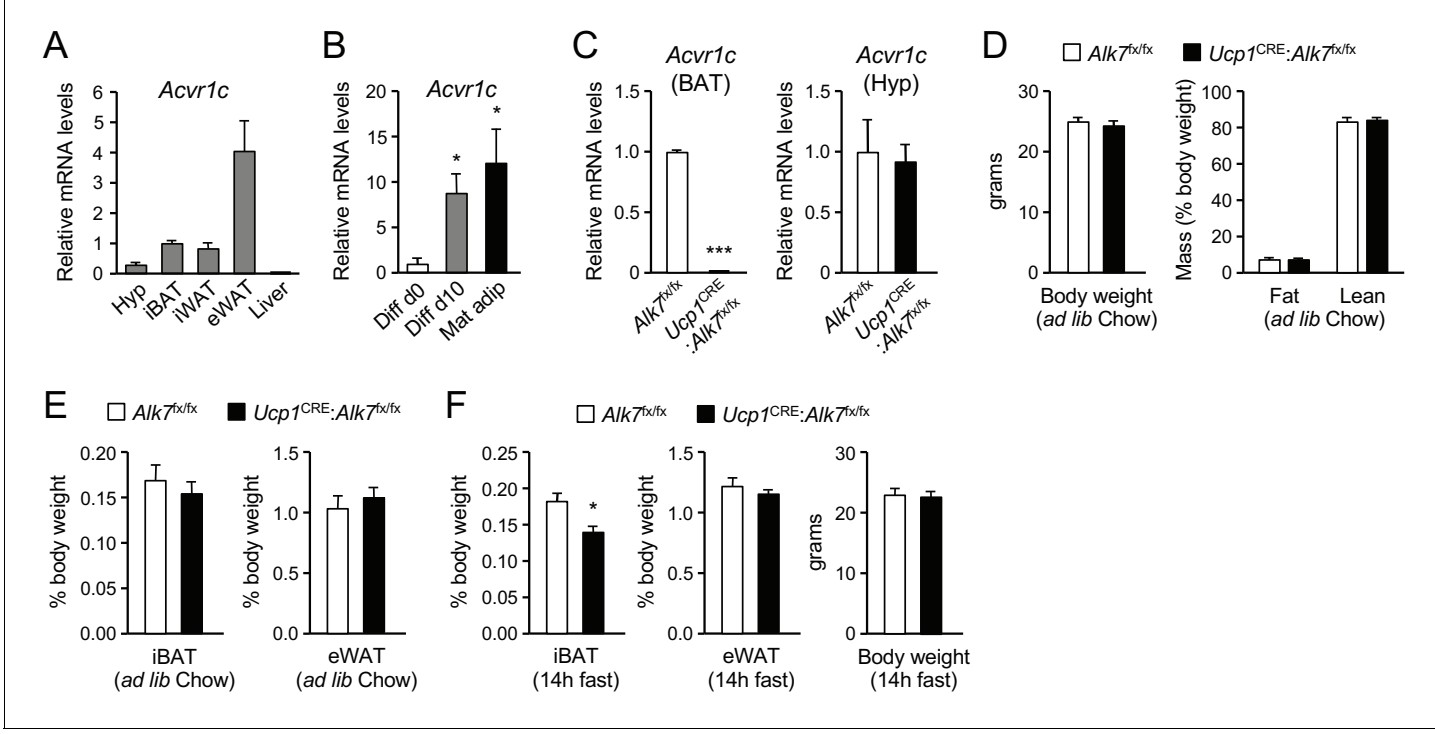

**Figure 1.** Reduced iBAT mass in $Ucp1^{CRE}$:$Alk7^{fx/fx}$ conditional mutant mice after fasting. (A) Q-PCR determination of $Acvr1c$ mRNA expression in hypothalamus (Hyp), interscapular BAT (iBAT), inguinal WAT (iWAT), epididymal WAT (eWAT) and liver of wild type mice. The values were normalized to mRNA levels in iBAT and are presented as average ± SEM. N = 4 or 6 (iWAT) mice per group. (B) Q-PCR determination of $Acvr1c$ mRNA expression in iBAT stromal vascular fraction (Diff d0), adipocytes differentiated in vitro (Diff d10), and freshly isolated mature adipocytes (Mat adip). The values were normalized to mRNA levels in the Diff d0 sample, and are presented as average ± SEM. N = 4 independent experiments. *, p<0.05; two-tailed Mann Whitney test. (C) Q-PCR determination of $Acvr1c$ mRNA expression in iBAT (left) and hypothalamus (right) from conditional mutant ($Ucp1^{CRE}$:$Alk7^{fx/fx}$) and control ($Alk7^{fx/fx}$) mice using primers flanking the kinase domain. The values were normalized to mRNA levels in control mice and are presented as average ± SEM. N = 4 mice per group. ***, p<0.001; two-tailed Mann Whitney test. (D) Body weight (left) at 2 months (ad libitum Chow diet) and fat and lean mass (expressed as percentage of body weight) assessed by magnetic resonance imaging (right). Values are presented as average ± SEM. N = 4 mice per group. (E) Relative iBAT and eWAT mass expressed as percentage of body weight at 2 months (ad libitum Chow diet) in conditional mutant and control mice. Values are presented as average ± SEM. N = 5 mice per group. (F) Relative iBAT mass, eWAT mass, and body weight at 2 months (ad libitum Chow diet) following 14 hr fasting in conditional mutant and control mice. Values are presented as average ± SEM. N = 5 mice per group. *, p<0.05; two-tailed unpaired Student's t-test.

The online version of this article includes the following figure supplement(s) for figure 1:

**Figure supplement 1.** Normal energy consumption and food intake in mutant mice lacking ALK7 in BAT.
**Figure supplement 2.** Normal BAT differentiation in mice lacking ALK7 in brown adipocytes.

mutant mice ($Ucp1^{CRE}$:$Alk7^{fx/fx}$), $Acvr1c$ mRNA expression in BAT was almost completely abolished (*Figure 1C*), confirming that ALK7 is exclusively expressed by brown adipocytes in this tissue. $Acvr1c$ expression was spared in other tissues, including hypothalamus (*Figure 1C*). At 2 months of age, $Ucp1^{CRE}$:$Alk7^{fx/fx}$ mutant mice showed body weight and fat composition indistinguishable from control mice under Chow diet (*Figure 1D*). Energy consumption was not affected by the lack of ALK7 in BAT, as demonstrated by normal oxygen consumption ($VO_2$) under both fed and fasted conditions (*Figure 1—figure supplement 1A*). Similarly, respiratory exchange ratio (RER), which reports the relative usage of carbohydrates and fat, and food intake were also comparable between $Ucp1^{CRE}$:$Alk7^{fx/fx}$ and control mice (*Figure 1—figure supplement 1B,C*). Both iBAT and eWAT mass relative to body weight were also normal in 2 month old mutants (*Figure 1E*), as well as expression of a battery of BAT differentiation and maturation markers and mitochondrial copy number (*Figure 1—figure supplement 2A,B*), indicating that ALK7 is not required by brown adipocytes for normal BAT development. Interestingly, overnight (14 hr) fasting induced a significant reduction in iBAT weight in 2 month old $Ucp1^{CRE}$:$Alk7^{fx/fx}$ mice, despite normal eWAT mass (*Figure 1F*).

Histological analysis of iBAT revealed a significant decrease in lipid droplet size in 2 month old *Ucp1*<sup>CRE</sup>:*Alk7*<sup>fx/fx</sup> mice compared to age-matched controls (*Figure 2A*). A similar reduction was observed in the iBAT of global *Alk7*<sup>-/-</sup> knock-out mice (*Figure 2—figure supplement 1A*). A proteomics analysis of iBAT lipid droplets from *Alk7*<sup>-/-</sup> knock-out mice revealed increased levels of adipose triglyceride lipase (ATGL), the rate-limiting enzyme of lipolysis, despite normal levels of other major lipid droplet proteins, including hormone-sensitive lipase (HSL) (*Figure 2B*). Total iBAT lysates from *Ucp1*<sup>CRE</sup>:*Alk7*<sup>fx/fx</sup> conditional mutant mice showed increased ATGL levels compared to controls (*Figure 2C*), although the difference did not reach statistical significance (p=0.077). However, ATGL protein levels were robustly increased in the mutant iBAT 14 hr after fasting (*Figure 2C*), which was in line with the reduced iBAT mass observed in fasted mutant animals (*Figure 1F*). ATGL levels were not changed by fasting in BAT of wild type mice (*Figure 2—figure supplement 1B*). No differences could be detected in the levels of phosphorylated HSL (P-HSL$^{S563}$, *Figure 2C*). In agreement with this, mRNA levels of *Adrb1* and *Adrb3*, encoding catecholamine receptors in adipocytes, were not significantly different between genotypes in either fed or fasted conditions (*Figure 2—figure supplement 1C,D*). Enhanced ATGL protein levels in the mutants were accompanied by a strong trend (p=0.078) towards increased *Atgl* mRNA levels after fasting (*Figure 2D*). iBAT from fasted conditional mutant mice also showed a significant decrease in the mRNA levels of G0/G1 switch gene 2 (*G0S2*), which encodes an inhibitor of ATGL activity (*Yang et al., 2010*; *Figure 2E*). In addition, we found elevated levels of C/EBPα (*Figure 2F*), a well known regulator of adipocyte differentiation which has been shown to collaborate with PPARγ to upregulate ATGL expression in adipocytes (*Yogosawa et al., 2013*; *Hasan et al., 2018*). The changes in ATGL and G0S2 levels prompted us to examine basal lipolysis in iBAT explants from *Ucp1*<sup>CRE</sup>:*Alk7*<sup>fx/fx</sup> and control mice as measured by basal glycerol release. Significantly increased levels of glycerol were detected in explant supernatants derived from fasted mutant mice compared to controls (*Figure 2G*), indicating abnormally enhanced basal lipolysis in iBAT of *Ucp1*<sup>CRE</sup>:*Alk7*<sup>fx/fx</sup> mice after fasting.

## Abnormally enhanced amino acid catabolism upon nutrient stress in iBAT lacking ALK7

The enhanced fat catabolism observed in iBAT lacking ALK7, particularly under fasting conditions, prompted us to investigate pathways involved in the regulation of metabolic balance. Insulin is a well known negative regulator of catabolic activity in adipose tissue during a postprandial state. Basal levels of AKT, a key downstream effector of insulin signaling, phosphorylated on Ser$^{473}$ (P-AKT$_{S473}$), which correlates with its activation state, were unchanged in iBAT of conditional mutant mice relative to controls, both under fed or fasted conditions (*Figure 2—figure supplement 2A*). P-AKT$_{S473}$ levels were neither changed by fasting in iBAT of wild type mice (*Figure 2—figure supplement 2B*). In addition, P-AKT$_{S473}$ levels were increased to the same extent in iBAT of both mutant and control mice in response to acute insulin administration (*Figure 2—figure supplement 2C*), indicating normal insulin sensitivity in iBAT lacking ALK7. A microarray analysis of iBAT from *Alk7*<sup>-/-</sup> global knock-out mice revealed several changes in gene expression in mutant iBAT compared to wild type controls (*Figure 3—figure supplement 1A,B*), including upregulation in the level of *Prodh* mRNA, encoding proline dehydrogenase (POX), a mitochondrial enzyme that catalyzes the first step in the degradation of proline, and a critical component of metabolic responses to nutrient stress in cancer cells (*Pandhare et al., 2009*; *Phang, 2019*). In line with this, *Prodh* mRNA levels were increased after 14 hr fasting in iBAT of both *Ucp1*<sup>CRE</sup>:*Alk7*<sup>fx/fx</sup> conditional mutant and control mice, but significantly more so in the mutants (*Figure 3A*). Similarly, mRNAs encoding enzymes involved in the degradation of alanine and branched amino acids, namely ALT1 and BCAT2, were also specifically upregulated upon fasting in the mutant iBAT (*Figure 3B*). Realizing that these three genes are all targets of KLF15, a key regulator of responses to nutritional stress in liver and skeletal muscle (*Gray et al., 2007*; *Haldar et al., 2012*), we examined the levels of *Klf15* mRNA in mutant and control iBAT. This revealed a significant induction of *Klf15* mRNA after fasting in iBAT from *Ucp1*<sup>CRE</sup>:*Alk7*<sup>fx/fx</sup> mutant mice, but not from control mice (*Figure 3C*). As expected, fasting induced *Klf15* mRNA in the liver regardless of genotype (*Figure 3C*). At the protein level, similar changes were observed for KLF15 (*Figure 3D*), ALT1, BCAT2 (*Figure 3E*) and POX (see below). In all cases, fasting induced a more pronounced increase in mutant BAT compared to control.

*Klf15* is one major target of glucocorticoid signaling, mediating several catabolic responses to glucocorticoids in liver and skeletal muscle (*Shimizu et al., 2011*; *Sasse et al., 2013*), but its role

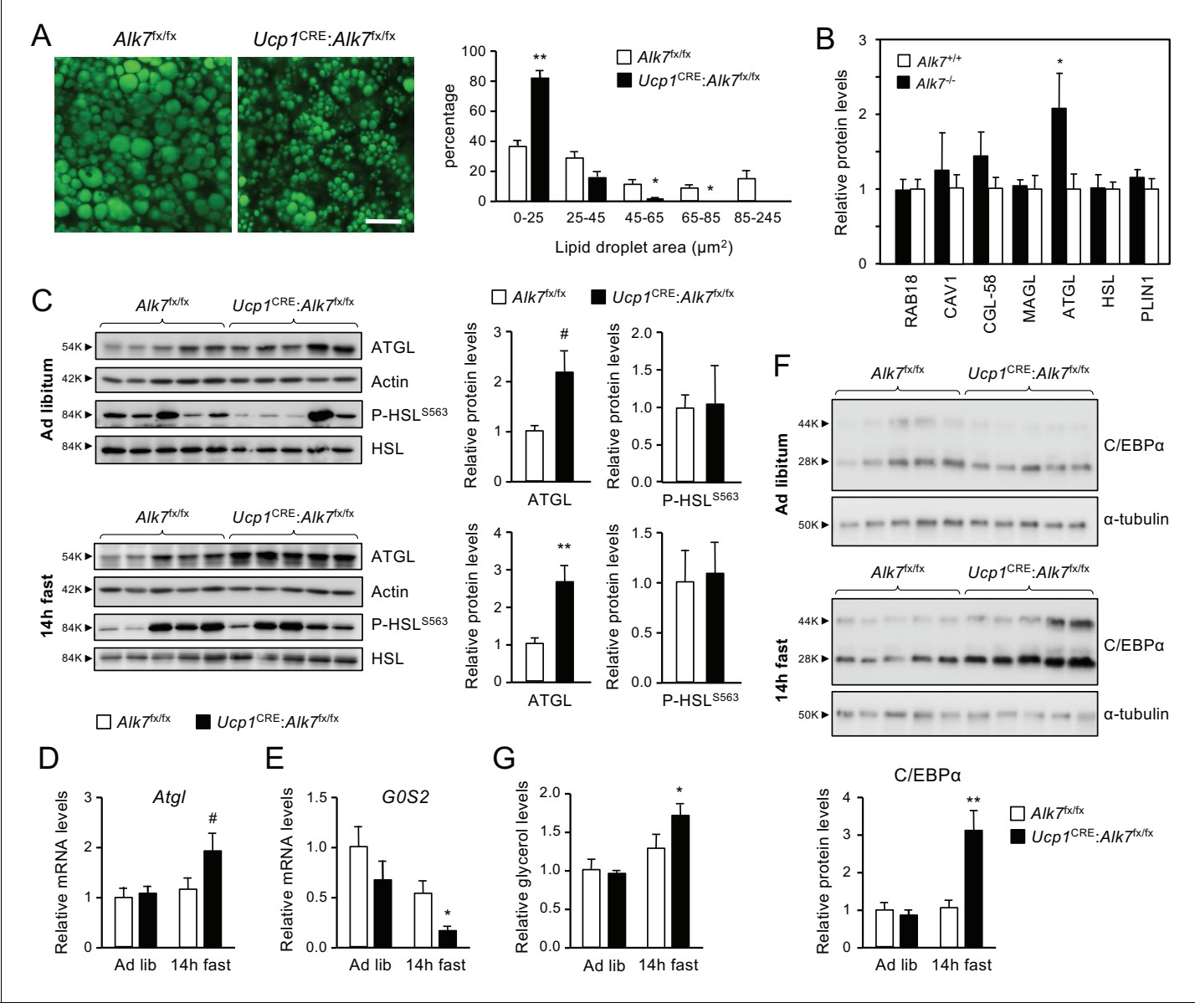

**Figure 2.** Fasting induces abnormally increased fat catabolism in BAT of *Ucp1*^CRE^:*Alk7*^fx/fx^ conditional mutant mice. (A) Representative BODIPY 493/503 staining of iBAT sections of conditional mutant (*Ucp1*^CRE^:*Alk7*^fx/fx^) and control (*Alk7*^fx/fx^) mice. Scale bar, 20 µm. Histograms to the right show quantitative analysis of lipid droplet size. Values are presented as average ± SEM. N = 4 mice per group. *, p<0.05; **, p<0.01; vs. control, respectively; unpaired Student's t-test. (B) Proteomic analysis of BAT lipid droplet fractions in global *Alk7*^-/-^ knock-out mice and wild type controls. PLIN1, Perilipin 1; HSL, Hormone-sensitive lipase; ATGL, Adipose triglyceride lipase; MAGL, Monoacylglycerol lipase; CGL-58, Comparative Gene Identification-58; CAV1, caveolin-1. Values are presented as average ± SEM. N = 4 mice per genotype. *, p<0.05, unpaired Student's t-test. (C) Western blot analysis of ATGL and phosphorylated HSL (P-HSL^S563^) in iBAT of 2 month old conditional mutant and control mice fed Chow ad libitum (D) or after 14 hr fasting (E). Histograms to the right show quantitative analyses of protein levels normalized to actin (for ATGL) or total HSL (for P-HSL^S563^) signals from re-probed blots, relative to those in control *Alk7*^fx/fx^ mice. N = 5 mice per genotype. #, p=0.077; *, p<0.05; two-tailed unpaired Student's t-test. (D, E) Q-PCR determination of *Atgl* (D) and *G0S2* (E) mRNA expression in iBAT of 2 month old conditional mutant and control mice fed Chow ad libitum or after 14 hr fasting. The values were normalized to mRNA levels in control mice fed ad libitum and are presented as average ± SEM. N = 4 mice per group. #, p=0.078 vs. control; *, p<0.05 vs. control; two-tailed unpaired Student's t-test. (F) Western blot analysis of C/EBPα in iBAT of 2 month old conditional mutant and control mice fed Chow ad libitum (D) or after 14 hr fasting (E). Histogram below show quantitative analyses of protein levels normalized to α-tubulin signals from re-probed blots, relative to those in control Alk7^fx/fx^ mice. C/EBPα runs as two bands of 28 and 44 kDa, respectively. N = 5 mice per genotype. **, p<0.01; two-way ANOVA. (G) Basal lipolysis measured as glycerol release ex vivo from iBAT explants from conditional mutant and control mice fed ad libitum or after 6 hr fasting. Values were normalized to ad libitum levels in control mice and are presented as average ± SEM. N = 6 (ad lib) and 5 (fast) mice per group. *, p<0.05 vs. control; two-tailed unpaired Student's t-test.

*Figure 2 continued on next page*

*Figure 2 continued*

The online version of this article includes the following figure supplement(s) for figure 2:

**Figure supplement 1.** Reduced lipid droplets in iBAT from Alk7 knock-out mice, unchanged levels of ATGL protein in fasted wild type mice, and normal Adrb1 and Adrb3 mRNA levels in iBAT lacking ALK7.
**Figure supplement 2.** Normal P-AKT levels and insulin sensitivity in iBAT lacking ALK7.

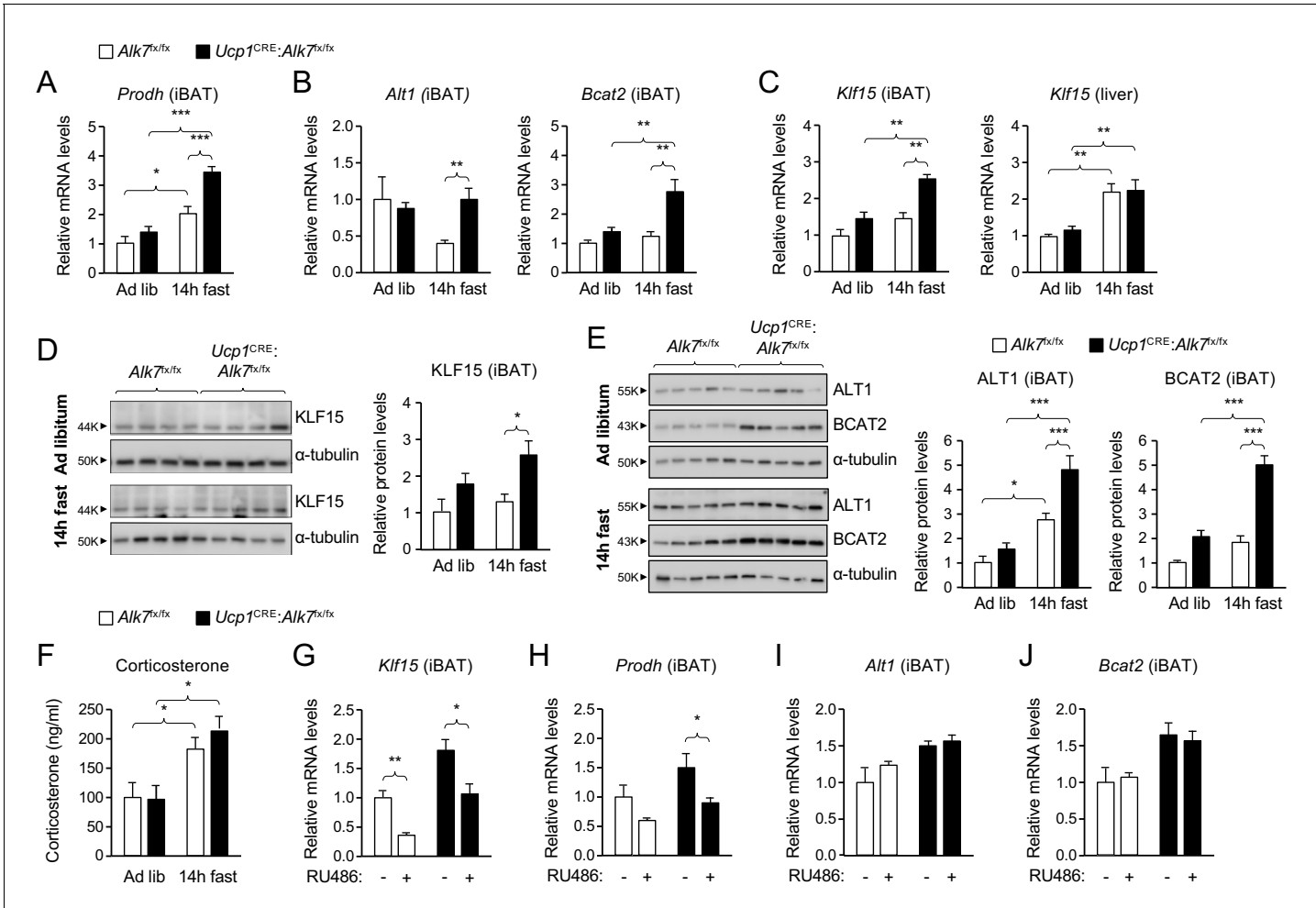

**Figure 3.** Abnormally enhanced amino acid catabolism upon nutrient stress in iBAT lacking ALK7. (A, B) Q-PCR determination of *Prodh* (A) and *Alt1* and *Bcat2* (B) mRNA expression in iBAT of 2 month old conditional mutant and control mice fed Chow ad libitum or after 14 hr fasting. The values were normalized to mRNA levels in control mice fed ad libitum and are presented as average ± SEM. N = 9 (A) or 5 (B) mice per group. *, p<0.05; ***, p<0.001; vs. control, respectively; unpaired Student's t-test. (C) Q-PCR determination of Klf15 mRNA expression in iBAT (left) and liver (right) of 2 month old conditional mutant and control mice fed Chow ad libitum or after 14 hr fasting. The values were normalized to mRNA levels in control mice fed ad libitum and are presented as average ± SEM. N = 5 mice per group. **, p<0.01; vs. control, respectively; two-way ANOVA with Bonferroni post-test. (D, E) Western blot analysis of KLF15, (D) ALT1, and BCAT2 (E) in iBAT of 2 month old conditional mutant and control mice fed Chow ad libitum or after 14 hr fasting, as indicated. Histograms show quantitative analyses of protein levels normalized to α-tubulin signals from re-probed blots, relative to control Alk7^fx/fx mice fed ad libitum. N = 3 experiments each in triplicate (mean ± SEM). *, p<0.05; ***, p<0.001; two-way ANOVA with Tukey's multiple comparisons test. (F) Serum corticosterone levels (ng/ml) in 2 month old conditional mutant and control mice fed Chow ad libitum or after 14 hr fasting. Values are presented as average ± SEM. N = 9 mice per group. *, p<0.05 vs. control; unpaired Student's t-test. (G–J) Q-PCR determination of *Klf15* (G), *Prodh* (H), *Alt1* (I) and *Bcat2* (J) mRNA expression in iBAT of 14h-fasted conditional mutant and control mice 4 hr after injection with RU486 or vehicle, as indicated. The values were normalized to mRNA levels in control mice injected with vehicle, and are presented as average ± SEM. N = 5 mice per group. *, p<0.05; **, p<0.01; vs. control, respectively; two-way ANOVA with Bonferroni post-test.

The online version of this article includes the following figure supplement(s) for figure 3:

**Figure supplement 1.** Microarray analysis of genes differentially expressed in iBAT of Alk7-/- global knock-out mice compared to wild type.

and regulation in BAT had not been investigated. We first verified that overnight fasting increased corticosterone levels in serum of both $Ucp1^{CRE}$:$Alk7^{fx/fx}$ mutants and control mice to a similar extent (*Figure 3F*). We then asked whether *Klf15* mRNA expression in iBAT was similarly sensitive to glucocorticoid signaling, as it has been demonstrated in other tissues. To this end, we administered the glucocorticoid receptor (GR) antagonist RU486 to conditional mutant and control mice 4 hr prior to the end of a 14 hr fasting period, after which iBAT was collected for mRNA analysis. We note that, in addition to function as a GR antagonist, RU486 has been shown to exert complex effects on a range of other steroid receptors, most notably as an antagonist of the progesterone receptor (*Lin et al., 2001*; *Zhang et al., 2006*). However, levels of progesterone in serum follow a circadian pattern, and reach a peak during night time in male rodents (such as the mice used here) (*Kalra and Kalra, 1977*). By administering RU486 in the morning, we believe that such effects were minimized in our studies. *Klf15* mRNA levels were reduced by RU486 treatment in both strains of mice (*Figure 3G*), indicating that *Klf15* expression is also under glucocorticoid regulation in BAT. Importantly, RU486 treatment also restored *Prodh* mRNA levels in iBAT of conditional mutant mice back to the level found in control mice (*Figure 3H*). Expression of *Alt1* and *Bcat2* mRNAs were not affected by RU486 treatment (*Figure 3I,J*), suggesting perhaps a lower sensitivity to KLF15 levels in these genes.

## Activin B suppresses expression of mRNAs encoding KLF15 and amino acid degrading enzymes in isolated mouse and human brown adipocytes

The results described above indicated that ALK7 signaling may negatively regulate the expression of genes involved in lipid and amino acid catabolism in brown adipocytes. In order to test this more directly, we established cultures of brown adipocytes derived by differentiation in vitro of iBAT SVF extracted from 2 month old wild type mice. After 10 days of differentiation, the expression of mRNAs for *Klf15*, *Prodh*, *Alt1* and *Bcat2* were strongly upregulated in these cultures compared to the levels in iBAT SVF cells (*Figure 4A*). Treatment with the ALK7 ligand activin B significantly reduced the expression of the four mRNAs in differentiated brown adipocytes (*Figure 4B*). This response was effectively suppressed by SB431542, an inhibitor of type I receptors for TGF-βs and activins, including ALK7 (*Inman et al., 2002*). Interestingly, SB431542 could on its own moderately increase the mRNA expression of the four genes, although the effect did not reach statistical significance (*Figure 4B*), perhaps reflecting the activities of endogenously produced ligands. Activin B had no effect on brown adipocytes lacking ALK7 (*Figure 4C*), indicating that its effects were mediated by the ALK7 receptor. In line with the reduced *G0S2* mRNA levels observed in iBAT from fasted $Ucp1^{CRE}$:$Alk7^{fx/fx}$ mutants (*Figure 2F*), treatment with activin B increased, while SB431542 decreased, expression of this gene in cultured brown adipocytes (*Figure 4D*). We detected a trend towards reduction of *Atgl* mRNA expression with activin B, and increased expression with SB431542, in agreement with the results in vivo, but these trends did not reach statistical significance. Regulation of *Atgl* mRNA expression by ALK7 may require additional, fasting-induced signals. Activin B had not significant effects on the mRNA levels of *Ucp1* or *Prdm16* (*Figure 4—figure supplement 1*). Importantly, similar observations could be made in cultured human brown adipocytes, in which activin B treatment also resulted in reduced levels of *Klf15* and *Prodh* mRNAs (*Figure 4E*). Together these results suggest that ALK7 signaling can directly suppress the expression of a series of mRNAs encoding diverse regulators of fat and amino acid catabolism.

## Increased proline-dependent ATP generation in mitochondria from iBAT lacking ALK7

The increased levels of *Prodh* mRNA, encoding the mitochondrial enzyme POX, in mutant iBAT prompted us to examine the levels of a battery of mitochondrial proteins, including POX, UCP1 and components of respiratory complexes I to V, in iBAT from $Ucp1^{CRE}$:$Alk7^{fx/fx}$ mutant and control mice, fed ad libitum as well as after 14 hr fasting. iBAT from mutant mice showed a moderate increase in POX protein levels compared to control mice (*Figure 5A*). However, fasting induced a significantly greater increase in POX protein levels in the mutant iBAT (*Figure 5A*). In addition, SDHA, a subunit of succinate dehydrogenase, was also differentially enhanced by fasting in the mutant iBAT (*Figure 5A*). Interestingly, POX and succinate dehydrogenase are both physically and functionally coupled in mitochondrial complex II (*Hancock et al., 2016*). None of the other

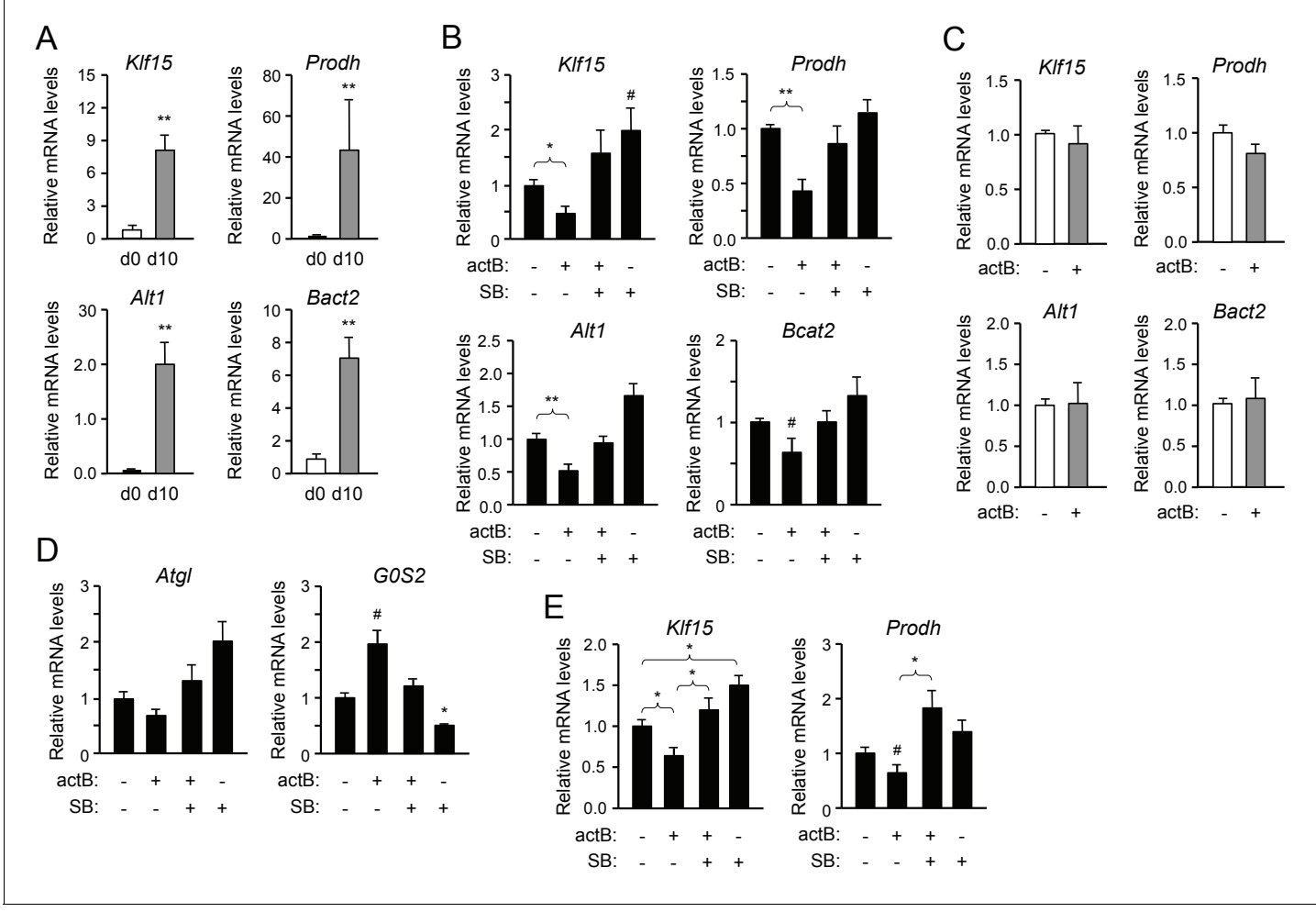

**Figure 4.** Activin B suppresses expression of mRNAs encoding KLF15 and amino acid degrading enzymes in isolated mouse and human brown adipocytes. (A) Q-PCR determination of *Klf15*, *Prodh*, *Alt1* and *Bcat2* mRNA expression in primary cultures of differentiated (d10) compared to non-differentiated (d0) brown adipocytes isolated from iBAT of wild type mice. The values were normalized to mRNA levels at d0 and are presented as average ± SEM. N = 5 independent experiments each performed in duplicate. **, p<0.01 vs. d0; unpaired Student's t-test. (B) Q-PCR determination of *Klf15, Prodh, Alt1* and *Bcat2* mRNA expression in primary cultures of differentiated brown adipocytes isolated from iBAT of wild type mice treated with activin B (actB) or SB-431542 (SB) for 24 hr as indicated. The values were normalized to levels in untreated cultures and are presented as average ± SEM. N = 5 independent experiments each performed in duplicate. #, p=0.088; *, p<0.05; **, p<0.01 vs. untreated; two-tailed paired Student's t-test. (C) Q-PCR determination of *Klf15, Prodh, Alt1* and *Bcat2* mRNA expression in primary cultures of differentiated brown adipocytes isolated from iBAT of *Alk7*[-/-] knock-out mice treated with activin B (actB) for 24 hr as indicated. The values were normalized to levels in untreated cultures and are presented as average ± SEM. N = 3 independent experiments each performed in duplicate. (D) Q-PCR determination of *Atgl* and *G0S2* mRNA expression in primary cultures of differentiated brown adipocytes isolated from iBAT of wild type mice treated with activin B (actB) or SB-431542 (SB) for 24 hr as indicated. The values were normalized to levels in untreated cultures and are presented as average ± SEM. N = 5 independent experiments each performed in duplicate. #, p=0.05; *, p<0.05 vs. untreated; two-tailed paired Student's t-test. (E) Q-PCR determination of *Klf15* and *Prodh* mRNA expression in primary cultures of differentiated human brown adipocytes isolated treated with activin B (actB) or SB-431542 (SB) for 24 hr as indicated. The values were normalized to levels in untreated cultures and are presented as average ± SEM. N = 5 independent experiments each performed in duplicate. *, p<0.05; vs. untreated; two-tailed unpaired Student's t-test.

The online version of this article includes the following figure supplement(s) for figure 4:

**Figure supplement 1.** Effect of activin B on mRNA expression of BAT markers Ucp1 and Prdm16 assessed in cultured brown adipocytes.

mitochondrial proteins investigated were found to be affected in the mutants, including UCP1, COXIV, Rieske FeS, NDUFA10 and beta F1 ATPase (*Figure 5—figure supplement 1A–D*). Levels of POX and UCP1 were not significantly changed by fasting in BAT of wild type mice (*Figure 5—figure supplement 1E*). Although we failed to detect any significant change in UCP1 expression upon treatment with activin B (*Figure 4—figure supplement 1*) or deletion of ALK7 in adult BAT

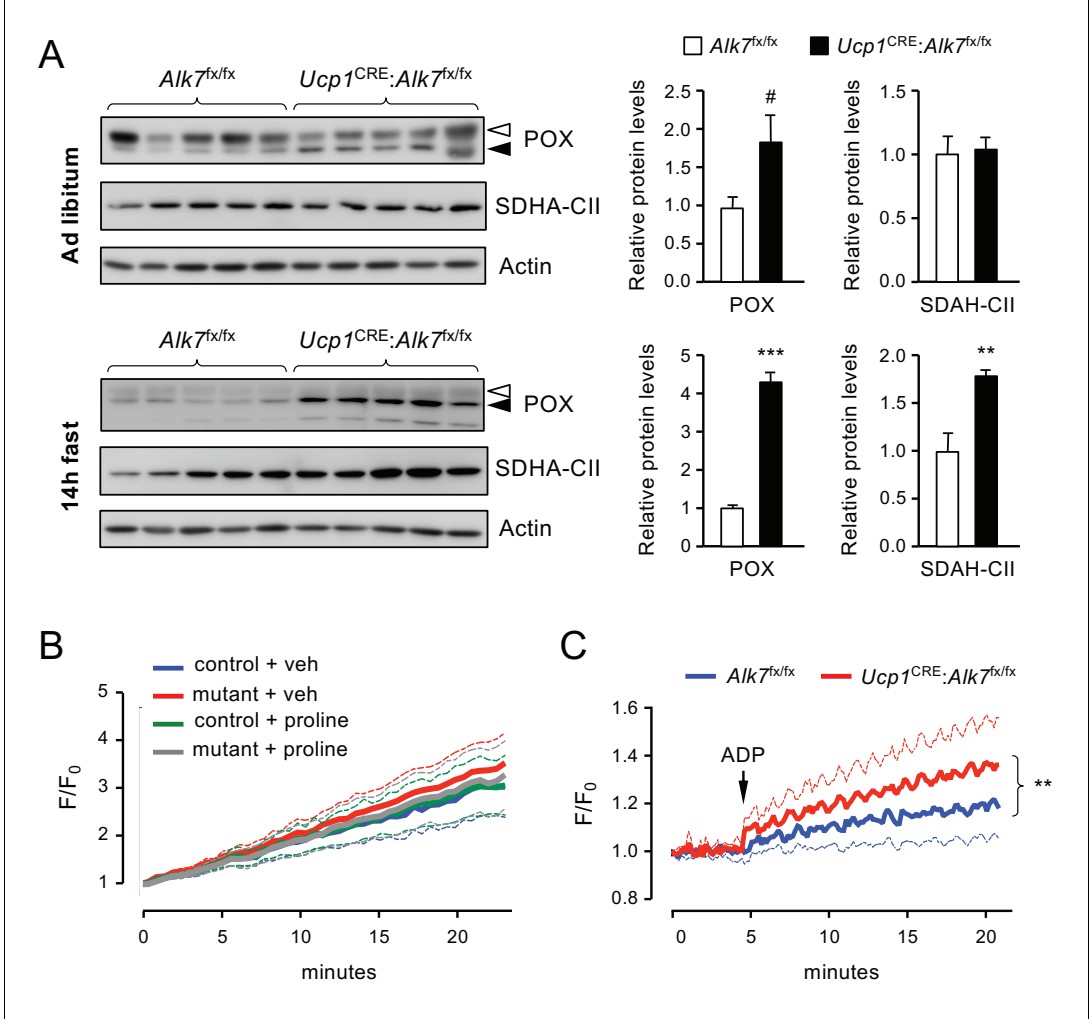

**Figure 5.** Increased proline-dependent ATP generation in mitochondria from iBAT lacking ALK7. (**A**) Western blot analysis of proline dehydrogenase (POX) and succinate dehydrogenase CII subunit (SDHA-CII) in iBAT of 2 month old conditional mutant and control mice fed Chow ad libitum (top) or after 14 hr fasting (bottom). Solid arrowheads point to POX band, open arrowheads denote unspecific band. Histograms to the right show quantitative analyses of protein levels normalized to actin signals from re-probed blots, relative to those in control *Alk7*fx/fx mice. N = 5 mice (mean ± SEM). #, p=0.069; **, p<0.01; ***, p<0.001; two-tailed unpaired Student's t-test. (**B**) Traces of ROS production in mitochondria isolated from iBAT of 14h-fasted conditional mutant (red and grey) and control (blue and green) mice after vehicle (blue and red) or proline stimulation (green and grey). Dotted lines represent standard error. N = 3 independent experiments. (**C**) Traces of proline-induced ATP production in mitochondria isolated from iBAT of 14h-fasted conditional mutant (red) and control (blue) mice. Dotted lines represent standard error. N = 3 independent experiments. **, p<0.01; two-way ANOVA.

The online version of this article includes the following figure supplement(s) for figure 5:

**Figure supplement 1.** Expression of mitochondrial proteins in BAT of control and conditional mutant mice fed ad libitum or after 14 hr fasting.

**Figure supplement 2.** Control experiments for measurements of ROS and ATP production.

---

(*Figure 5—figure supplement 1A*), a previous study using cultured wild type brown adipocytes reported reduced UCP1 expression during early differentiation and a modest increase in mature adipocytes following stimulation with activin AB, which the authors attributed to ALK7 signaling (*Balkow et al., 2015*). We note that all activins are also able to activate the related receptor ALK4, which is highly expressed in adipocyte precursors and also to some degree in mature cells, therefore such changes, as well as their possible physiological relevance, can not be attributed to ALK7 without evidence from loss-of-function studies.

We investigated possible functional consequences of the increased levels of POX in BAT of fasted *Ucp1*CRE:*Alk7*fx/fx mutant mice by assessing two of the most important effects attributed to proline

oxidation by POX, namely production of reactive oxygen species (ROS) (*Donald et al., 2001*; *Zarse et al., 2012*; *Goncalves et al., 2014*) and generation of ATP (*Pandhare et al., 2009*; *Phang, 2019*). In control experiments, rotenone, a well known inducer of ROS production, produced a robust increase in ROS levels in BAT mitochondria (*Figure 5—figure supplement 2A*). However, in our hands, addition of proline to mitochondria isolated from iBAT of fasted mice failed to induce ROS production, regardless of genotype (*Figure 5B*). Next, we assessed ATP synthesis in BAT mitochondrial fractions from 14 hr fasted mutant and control mice. Compared to its effects on liver mitochondria (*Figure 5—figure supplement 2B*), proline supported a modest increase in ATP production in mitochondria isolated from BAT of fasted control mice (*Figure 5C*). However, ATP generation was significantly elevated in BAT mitochondria of fasted *Ucp1*[CRE]:*Alk7*[fx/fx] mutant mice compared to controls (*Figure 5C*). These results suggest that nutrient stress induces elevated levels of POX which in turn lead to enhanced ATP generation in mitochondria from BAT lacking ALK7.

## Fasting-induced hypothermia in mice lacking ALK7 in brown adipocytes

In order to test the possible physiological significance of the changes observed in the metabolic functions of mutant iBAT lacking ALK7 we exposed control and mutant animals to acute cold (5°C for 4 hr) in metabolic cages and assessed energy consumption, body temperature and activity during this period. A second group of mice was fasted for 14 hr prior to cold exposure to test the effects of more stringent nutritional conditions. The body weights of mice placed in metabolic chambers was not different between genotypes. In fed mice, a 4 hr cold exposure did not reveal any alterations in either $VO_2$, body temperature, RER (*Figure 6A to C*) or activity (*Figure 6—figure supplement 1A*) in the mutant mice compared to controls. The $VO_2$ traces showed an expected increase during the first hour of exposure to cold, followed by a plateau that was largely maintained in the two genotypes (*Figure 6A*). However, significant changes were observed in animals that had been fasted for 14 hr prior to cold exposure. Mutant animals that had been fasted were unable to keep the higher levels of $VO_2$ observed in control animals (*Figure 6D*) and, as a consequence, displayed a very rapid drop in body temperature (*Figure 6E*), with over 70% of animals showing temperatures lower than 32°C after 3 hr (compared to 20% of controls). Overall activity was similar to controls in the mutant mice during this period (*Figure 6—figure supplement 1B*). In these conditions, both genotypes showed a similar switch to usage of free-fatty acids (FFA) as energy source, as shown by RER close to 0.7 (*Figure 6F*), in line with the low levels of serum glucose and insulin observed upon cold exposure, which did not show differences between genotypes (*Figure 6—figure supplement 1C,D*). Both mutants and controls also showed similar serum levels of FFA and triglycerides (*Figure 6—figure supplement 1E,F*), indicating normal uptake of circulating lipids in the mutants (abnormal uptake is known to result in elevated levels in serum). We note that mutant mice exposed to 5°C for 8 hr with unrestricted access to food showed no difference in body temperature compared to controls (*Figure 6—figure supplement 1G*), indicating that the deficit in the mutants is specific to situations of nutritional stress. Lastly, we tested whether responses to norepinephrine (NE), a key activator of BAT upon cold exposure, were affected in the conditional mutants lacking ALK7 in brown adipocytes. NE content and turnover was comparable in iBAT from fasted mutant and control mice during cold exposure (*Figure 6—figure supplement 2A*). Also, *Ucp1* mRNA was similarly induced by cold exposure in both genotypes (*Figure 6—figure supplement 2B*), and levels of activated mTORC2 (phosphorylated at Ser[2481]) and HSL (phosphorylated at Ser[563]), which are targets of NE signaling in BAT (*Albert et al., 2016*), were similar in fasted mutant and control mice upon cold exposure (*Figure 6—figure supplement 2C,D*), suggesting normal NE signaling in mutant BAT. Together, these results suggest that hypothermia in mice lacking ALK7 in BAT may be the consequence of abnormally high catabolism of fat and amino acids upon nutrient stress, resulting in depletion of energy depots necessary to defend body temperature upon cold exposure.

In the final set of studies, we asked whether the metabolic defects underlying the inability of the mutants to maintain body temperature upon acute cold exposure in conditions of nutrient stress, could lead to impaired cold adaptation upon prolonged exposure to lower temperatures, even with unrestricted access to food. Control and mutant animals of matched age and weights were first cold adapted for 14 days at 18°C, and then placed at 10°C for 21 days. While control animals maintained body temperature throughout the experiment, a significant drop in body temperature was observed in the mutants during the last week of the treatment (*Figure 6G*). This was accompanied by a decrease in relative iBAT weight in the mutants exposed to cold, similar to that observed upon

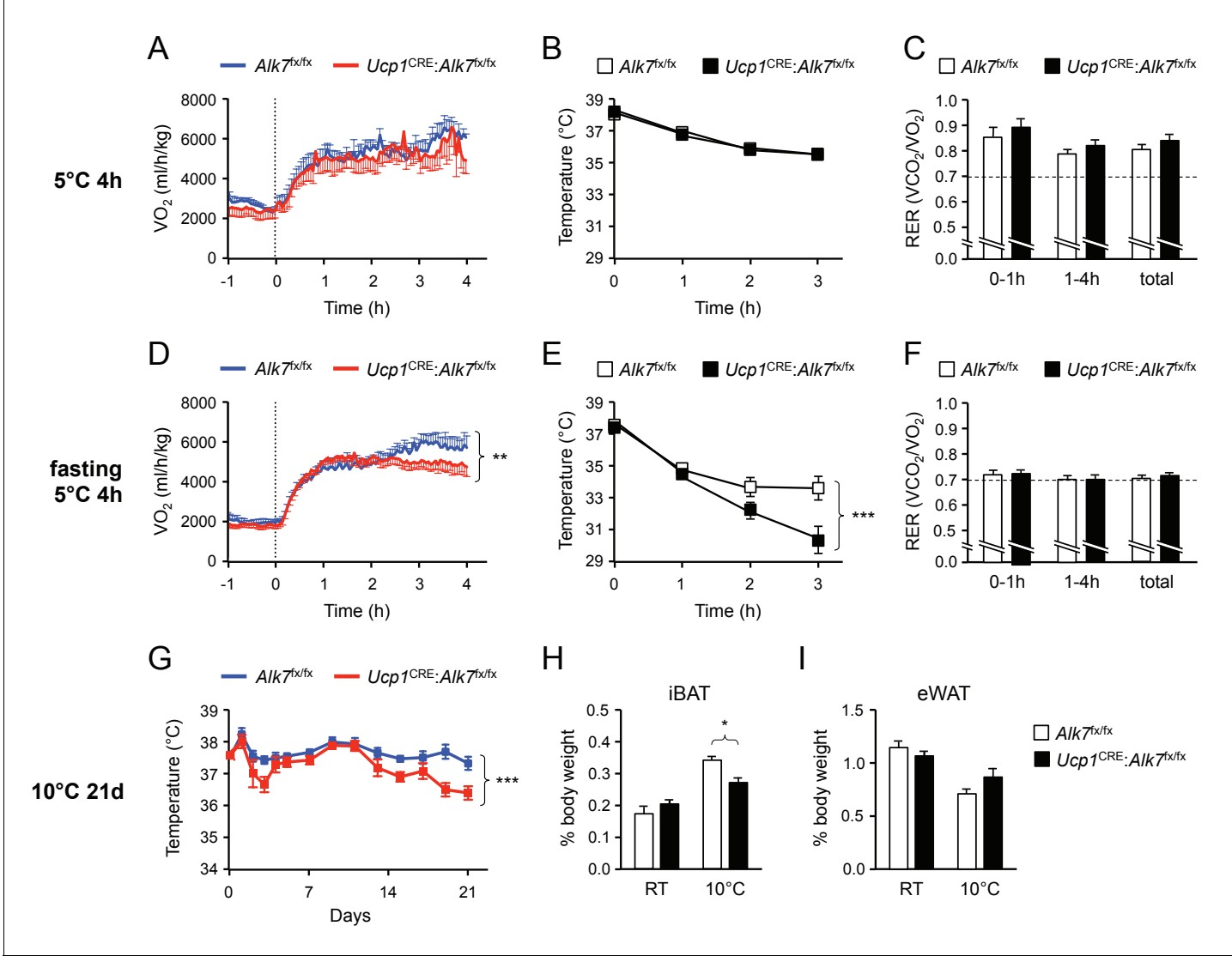

**Figure 6.** Fasting-induced hypothermia in mice lacking ALK7 in brown adipocytes. (**A, D**) $VO_2$ measured by indirect calorimetry in ad libitum fed (**A**) or 14h-fasted (**D**) 2 month old conditional mutant and control mice during exposure to 5℃ in metabolic cages. N = 5 (**A**) and 10–12 (**D**) mice per group. **, p<0.01; two-way ANOVA (**B, E**) Rectal temperature in ad libitum fed (**B**) or 14h-fasted (**D**) 2 month old conditional mutant and control mice during exposure to 5℃. N = 8 (**B**) and 10 (**E**) mice per group. ***, p<0.001; two-way ANOVA. (**C, F**) Respiratory exchange ratio (RER) measured by indirect calorimetry in ad libitum fed (**C**) or 14h-fasted (**F**) 2 month old conditional mutant and control mice during exposure to 5℃ in metabolic cages. N = 5 (**C**) and 10–12 (**F**) mice per group. (**G**) Rectal temperature in conditional mutant and control mice during chronic exposure to 10℃ for 21 days (preceded by 14d acclimatization at 18℃). Temperature was measured every 2 days during the light phase of the day cycle. N = 5 mice per group. ***, p<0.001; two-way ANOVA. (**H, I**) Relative iBAT and eWAT mass expressed as percentage of body weight in conditional mutant and control mice after 21d chronic cold exposure at 10℃. Values are presented as average ± SEM. N = 5 mice per group. *, p<0.05; one-way ANOVA with Tukey's post test. The online version of this article includes the following figure supplement(s) for figure 6:

**Figure supplement 1.** Influence of fasting and cold exposure on levels of glucose, insulin, activity, free-fatty acids, triglycerides and body temperature.
**Figure supplement 2.** Unchanged norepinephrine signaling in iBAT of Ucp1CRE:Alk7fx/fx mutant mice after fasting and acute cold exposure.
**Figure supplement 3.** Body weight, food intake and Ucp1 mRNA levels after chronic cold exposure (21 d at 10℃) in control and mutant mice.
**Figure supplement 4.** Expression of mRNA levels encoding thermogenic markers in inguinal WAT after chronic cold exposure (21 d at 10℃) in control and conditional mutant mice.
**Figure supplement 5.** Schematic model for possible signaling pathway related to the role of ALK7 in mediating BAT responses to nutrient stress.

fasting, but not in eWAT weight (*Figure 6H and I*). Body weight and food intake were comparable in the two genotypes after cold exposure (*Figure 6—figure supplement 3A,B*). Also *Ucp1* mRNA was induced to similar levels by prolonged cold treatment in BAT of control and mutant mice (*Figure 6—figure supplement 3C*). Chronic cold exposure also induces *Ucp1* mRNA expression in subcutaneous WAT depots, particularly iWAT, a processed generally known as 'browning'. We therefore examined expression of several thermogenic markers in iWAT of control and *Ucp1*^CRE^: *Alk7*^fx/fx^ mutant mice after prolonged cold exposure, but found that they were induced to comparable levels in both genotypes (*Figure 6—figure supplement 4A–E*), indicating normal browning of iWAT in conditional mutant mice after cold exposure. These results indicate that *Ucp1*^CRE^:*Alk7*^fx/fx^ mutant mice are inadequately adapted to prolonged cold exposure, perhaps due to premature depletion of energy depots in BAT, even in conditions of unrestricted food access.

## Discussion

Fasting triggers a range of catabolic activities that enable the usage of endogenous energy reservoirs, such as fat and proteins, allowing tissues to cope with their metabolic needs, ultimately contributing to the survival of the organism. Unlike liver, muscle and other tissues, our understanding of the physiological signals that adapt BAT function to different nutritional states is very limited. Based on the results of the present study, we propose that ALK7 functions to dampen catabolic activities triggered in BAT upon limited nutrient availability. Under nutrient stress, these catabolic functions, for example lipolysis and amino acid degradation, become abnormally amplified in brown adipocytes lacking ALK7, leaving the tissue energetically unable to cope with the demands imposed by low ambient temperatures. Recent studies have reported that circulating energy substrates are sufficient to fuel non-shivering thermogenesis under conditions that blunt BAT lipolysis, even upon acute cold exposure in the absence of food (*Schreiber et al., 2017*; *Shin et al., 2017*), leading to the notion that lipid droplet lipolysis in brown adipocytes is not essential for cold-induced thermogenesis regardless of food availability (*Shin et al., 2017*). However, these studies left open the question of the importance of endogenous BAT stores when animals confront lower temperatures after a previous period of prolonged fasting. Under such more stringent nutrient conditions, which we presume not to be uncommon in the wild, our findings indicate that energy reservoirs within BAT become crucially important to maintain body temperature. Our study indicates that abnormal catabolic function in BAT can indeed compromise responses to cold exposure.

Fasting induces expression of *Klf15* in liver, where it upregulates gluconeogenesis, and in muscle, where it promotes amino acid degradation, thereby providing precursors for liver gluconeogenesis at the expense of muscle protein (*Yamamoto et al., 2004*; *Gray et al., 2007*; *Shimizu et al., 2011*). Glucocorticoid signaling appears to be responsible for *Klf15* induction upon fasting in these tissues (*Gray et al., 2007*; *Sasse et al., 2013*). In contrast, *Klf15* expression was recently reported to be decreased in white adipocytes from fasted mice compared to fed mice (*Matoba et al., 2017*). The same study showed that WAT KLF15 inhibits lipolysis and promotes lipid storage in response to insulin, indicative of anabolic functions in this tissue. On the other hand, the regulation and possible functions of KLF15 in BAT have been unknown. Unlike WAT, and in line with observations made in liver and muscle, we find that fasting induces *Klf15* gene expression in BAT, which, suppression by RU486, suggests it to also be under the control of glucocorticoid signaling. Interestingly, muscle and BAT have been shown to originate from a common set of dermomyotome-derived precursor cells, which are distinct from those that give rise to WAT (*Wang and Seale, 2016*), suggesting a possible reason for their sharing a similar mode of *Klf15* regulation upon fasting. A recent study investigated BAT-specific GR knock-out mice and concluded that glucocorticoid signaling is dispensable for control of energy homeostasis in BAT (*Glantschnig et al., 2019*). However, this report did not examine the molecular components of the fasting response known to be regulated by glucocorticoids, including KLF15, nor were responses to cold exposure tested in fasted mice. Based on our results, we would anticipate significant changes under such conditions.

Previous studies in cancer cells have shown that POX can promote cell survival under conditions of limited nutrient availability through its ability to catabolize collagen-derived proline (*Olivares et al., 2017*). Proline is the most abundant amino acid in collagen, itself a major component of the extracellular matrix of many tissues, including BAT. Under conditions of low fatty acid and glucose availability, increased POX expression in BAT lacking ALK7 may lead to channeling of

TCA intermediates and mitochondrial generation of ATP, as observed in fasted mutant mice, leaving these animals with fewer reserves to successfully confront exposure to cold temperatures. In line with this, the body temperature of fasted mutant mice dropped significantly after 3 hr cold exposure, in parallel with lower energy expenditure. In line with our findings, a recent study reported that, upon cold exposure, BAT utilizes branched-chain amino acids for thermogenesis, promoting systemic clearance of these amino acids in both in mice and humans (*Yoneshiro et al., 2019*). The mitochondrial enzyme BCAT2 was found to be particularly important in this process. Although fasting had only a modest effect on the induction of BCAT2 in wild type mice, we found a 2.5-fold increase in mutant mice lacking ALK7 in BAT. Together, these observations suggest that premature depletion of branched-chain amino acids under nutrient stress predisposes mutant mice lacking ALK7 in BAT to succumb to acute cold exposure.

Our study reveals a novel mechanism for the role of ALK7 in mediating BAT responses to nutrient stress involving repression of the glucocorticoid/KL15/POX axis (*Figure 6—figure supplement 5*), which is distinct from the established role of ALK7 in regulating adrenergic activity in WAT during high caloric intake (*Guo et al., 2014*). Fasting is a well known stimulus of glucocorticoid release (*Dallman et al., 1993*), and through the GR, glucocorticoids can stimulate transcription of the *Klf15* gene in liver and muscle cells (*Gray et al., 2007*; *Sasse et al., 2013*). KLF15 in turn increases expression of amino acid degrading enzymes POX, ALT1, BCAT2 (*Gray et al., 2007*; *Shimizu et al., 2011*; *Sasse et al., 2013*). In addition, activation of GR signaling by dexamethasone has been shown to increase expression of C/EBPα in hepatoma cells (*Cram et al., 1998*) and the GR itself can interact directly with C/EBPα and increase its activity in different cell types (*Rüdiger et al., 2002*; *Muratcioglu et al., 2015*). C/EBPα is a well known regulator of adipocyte differentiation, and has been shown to collaborate with PPARγ to upregulate ATGL expression in adipocytes (*Yogosawa et al., 2013*; *Hasan et al., 2018*). It has also been demonstrated that KLF15 can regulate C/EBPα expression in adipocytes (*Asada et al., 2011*), establishing a further link between GR signaling and ATGL induction in these cells. Our results show that loss of ALK7 leads to amplification of these catabolic pathways, suggesting that ALK7 normally functions to suppress their activity (*Figure 6—figure supplement 5*). Although the detailed molecular mechanism remains to be clarified, our results suggest that ALK7 activation of Smad2/3, key mediators of signaling by TGF-β superfamily receptors, interferes with glucocorticoid signaling in adipocytes, either through direct binding of Smad proteins to the GR, or through intermediate proteins, such as transcriptional co-factors. In addition, Smad3 has been shown to interact directly with C/EBPβ and C/EBPδ, thereby inhibiting their ability to sustain C/EBPα expression in adipocytes (*Choy and Derynck, 2003*). Our finding of increased levels of C/EBPα in iBAT of mutant mice after fasting is in agreement with this observation, and suggests another possible route for the effects of ALK7 on ATGL expression.

Mutant mice lacking ALK7 in BAT were unable to maintain normal BAT mass and body temperature under chronic cold exposure (3 weeks), despite unrestricted access to food. We note that patients undergoing chronic (2 weeks) glucocorticoid treatment showed reduced BAT mass compared to controls (*Ramage et al., 2016*). It is possible that chronic exposure to overactive glucocorticoid-regulated pathways in BAT of conditional mutant mice may exaggerate BAT catabolic functions, resulting in reduced BAT mass and lower thermogenic performance, even under normal feeding conditions. At any rate, these observations indicate an important role for BAT ALK7 in the adaptation to chronic cold exposure.

In summary, we have discovered a novel role for the TGF-β superfamily receptor ALK7 in the adaptation of BAT physiology to variations in nutritional status, to our knowledge, the first mechanism described to regulate this important process. We find that ALK7 functions by limiting nutrient stress-induced overactivation of catabolic activities in brown adipocytes. Mechanistically, it does so by suppressing the levels of ATGL, a key enzyme for lipolysis, and amino acid degrading enzymes, including POX, through a novel pathway involving downstream effectors of glucocorticoid signaling, such as KLF15, a master regulator of fasting responses. Based on our results, we propose that ALK7 functions as a sensor to allow the brown adipocyte to adapt metabolically to the availability of nutrients. A better understanding of the mechanisms by which BAT function adjusts to fluctuations in nutrient availability will be critical for the development of safe methods to harness energy expenditure in BAT to combat human obesity and metabolic syndrome.

## Materials and methods

### Animals

Mice were housed under a 12 hr light-dark cycle, and fed a standard chow diet or a high-fat diet (HFD, 60% of calorie from fat; ResearchDiet D12492). The mouse lines utilized in this study have been described previously and are as follows: (i) conditional $Alk7^{fx/fx}$ (*Guo et al., 2014*); (ii) global knock-out $Alk7^{-/-}$ (*Jörnvall et al., 2004*); and (iii) CRE line $Ucp1^{CRE}$ (*Kong et al., 2014*); all back-crossed for at least 10 generations to a C57BL/6J background (considered as wild type). Animal protocols were approved by Stockholms Norra Djurförsöksetiska Nämnd (Stockholm North Ethical Committee for Animal Research) and are in accordance with the ethical guidelines of the Karolinska Institute.

### Cold exposure, calorimetry and body composition

For cold exposure, animals were housed in a Memmert HPP750 climate chamber at the indicated temperatures. For acute cold exposure, body temperature was measured every hour using a rectal thermometer. After 3 hr at 5°C, animals were sacrificed and iBAT extracted for molecular studies. For chronic cold exposure, animals were acclimatized to 18°C for 14 days prior to 21d exposure to 10°C. Body temperature was measured every 2 days during the light phase of the day cycle. Indirect calorimetry, food intake, and locomotor activity were assessed with a PhenoMaster Automatic Home Cage system (TSE Systems). Mice were housed individually with ad libitum access to food and water. Mice were acclimatized to the metabolic cages prior to automated recordings. For acute cold exposure in metabolic cages, mice were placed 4 hr at 5°C. Fat and lean mass were measured using a body composition analyzer EchoMRI-100TM.

### Insulin sensitivity, RU486 treatment and NE turnover

For insulin sensitivity test, mice fasted for 4 hr received an intraperitoneal injection of 0.75 U/kg Humulin-R insulin (Eli Lilly) or saline (vehicle). Mice were sacrificed 10 min later, and tissue samples were snap-frozen at −80°C for subsequent analysis. Ru486 (Sigma-Aldrich) was freshly formulated in DMSO before use. Weight-paired mice were fasted for 10 hr and then injected intraperitoneally with 5 mg of RU486 in 50 µl of DMSO or vehicle (only DMSO). Mice were sacrificed 4 hr later, and tissue samples were snap-frozen at −80°C for subsequent analysis. For analysis of NE turnover, mice fasted overnight (14 hr) were injected with 250 mg/kg of the norepinephrine synthesis blocker α-Methyl-DL-tyrosine methyl ester hydrochloride (AMT, Sigma-Aldrich) and immediately placed at 5°C. Mice were sacrificed after 3 hr cold exposure and tissue samples were snap-frozen at −80°C for subsequent NE quantification. Tissue NE was measured by ELISA kit (Labor Diagnostika Nord) according to the manufacturer's recommendation.

### Measurements in blood and serum samples

Blood samples were obtained by tail tip bleeding in the morning. Blood glucose was determined using a glucometer (Accutrend; Roche). Serum from blood was obtained by centrifugation of blood at 9391x;g for 15 min at 4°C. The pellet was discarded and serum samples were stored at −80° C for further analysis. Free fatty acid and triglyceride levels were measured in serum using the colorimetric quantification kits Half-micro test (Roche) and Infinity (ThermoScientific), respectively. Insulin was measured in serum with an ultra-sensitive mouse insulin ELISA kit (Mercodia). Corticosterone in serum was measured by ELISA kit (Enzo). All commercial kits were used following the manufacturer's recommendations.

### Ex vivo lipolysis

For ex vivo lipolysis, iBAT was dissected from mice that were either fed ad libitum or fasted for 6 hr, and tissue pieces were placed in DMEM at 37°C for 1 hr. The medium was then replaced with DMEM containing 2% fatty-acid-free BSA for 1 hr at 37°C. After incubation, the medium was collected and iBAT pieces were solubilized in 0.3N NaOH/0.1% SDS at 65°C overnight and subsequently centrifuged at 845x;g for 15 min at 4°C to remove the layer of fat. Protein content was determined using Pierce BCA Protein assay (Pierce, Thermo Fisher Scientific). Glycerol release to the

media was measured using a free glycerol reagent (Sigma-Aldrich), and normalized to the total amount of protein in the tissue samples.

Histological analysis of iBAT sections iBAT tissue samples were fixed in 4% PFA, embedded in 4% agar and cut into 50 µm-thick sections in a vibratome. Sections were incubated in 10% sucrose for 15 min, followed by 30% sucrose for 3 hr at 4°C, and permeabilization by four consecutive freeze-thaw cycles in 30% sucrose. Sections were stained with 10 µg/ml BODIPY 493/503 and mounted onto glass slides for imaging a confocal microscope (Zeiss). The area individual lipid droplets was measured with Zen software (Zeiss) and used for quantification of lipid droplet size.

## Isolation of mitochondria and measurements of ATP synthesis and ROS production

For mitochondria isolation, iBAT and liver were dissected out, washed and minced in BAT (250 mM sucrose and 0.1% fatty-acid-free BSA) or liver (125 mM sucrose, and 0.1% fatty-acid-free BSA) isolation buffer, respectively. The tissue was homogenized in a glass homogenizer in isolation buffer supplemented with protease inhibitors (Roche). Homogenates were filtered through a 70 µm mesh and nuclei were removed by centrifugation at 845xg for 10 min at 4°C in a microcentrifuge. Mitochondrial fractions were then collected by centrifugation at 9391xg for 15 min at 4°C, resuspended in MSK buffer (75 mM mannitol, 25 mM sucrose, 5 mM potassium phosphate, 20 mM Tris-HCl, 0.5 mM EDTA, 100 mM KCl, and 0.5% fatty-acid-free BSA, pH 7.4) and kept on ice until used. Proteins were measured by the BCA method.

ATP synthesis was determined fluorometrically in isolated mitochondria using a coupled enzyme assay with continuous monitoring of the reduction of NADP as described previously (*Korge et al., 2003*), with minor modifications. Mitochondrial fractions (0.5 mg/ml) were suspended in 150 µl of MSK buffer in the presence of 5 mM proline, 1 mM NADP, 10 mM glucose, 10 U/ml hexokinase, 5 U/ml glucose-6-P dehydrogenase, and 0.5% fatty-acid-free BSA. ATP synthesis was started by the addition of 100 µM ADP and measured as an increase in NADPH fluorescence (excitation = 340 nm, emission = 450 nm) at 37°C under constant stirring in a SpectraMax M2 microplate reader (Molecular Devices).

Mitochondrial ROS was measured as described previously with slight modifications (*Goncalves et al., 2014*). Mitochondrial fractions (0.01 mg/ml) were suspended in 0.15 ml of MSK buffer supplemented with 0.5% fatty-acid-free BSA in the presence of 12 U/ml horseradish peroxidase, 45 U/ml superoxide dismutase, and 50 µM Amplex UltraRed. Superoxide production was started by the addition of 5 mM proline and converted to $H_2O_2$ by superoxide dismutase. Appearance of $H_2O_2$ was monitored as the increase in fluorescence of the oxidized form of Amplex UltraRed (excitation = 545 nm, emission = 600 nm) at 37°C under constant stirring in a SpectraMax M2 microplate reader (Molecular Devices).

## Isolation and mass spectrometry analysis of lipid droplets

Lipid droplets were isolated and delipidated as described previously (*Brasaemle and Wolins, 2016*) with slight modifications. Tissue was placed in ice-cold hypotonic lysis medium (HLM, 10 mM Tris, pH 7.4, 1 mM EDTA) in the presence of a protease inhibitor cocktail (Roche), minced and homogenized by 20 strokes in a glass homogenizer. Lysates ($\approx$ 11 ml) were centrifuged at 26,000x;g for 30 min at 4°C in a SW41Ti rotor (Beckman). The floating lipid droplet layers were harvested with a glass pipette and adjusted to 25% sucrose and 100 mM sodium carbonate, pH 11.5, using 60% sucrose and 1M sodium carbonate stock solutions containing protease inhibitors, by gentle mixing by pipetting. These fractions ($\approx$ 4 ml) were layered into centrifuge tubes containing 1 ml cushions of 60% sucrose and then overlaid with $\approx$ 5 ml of 100 mM sodium carbonate, pH 11.5, with protease inhibitors followed by $\approx$ 1 ml of hypotonic lysis medium with protease inhibitors. Tubes were centrifuged at 26,000x;g for 30 min at 4°C in a SW41Ti rotor. Floating lipid droplets were harvested using a pipette tip into 2 ml microcentrifuge tubes. Residual carbonate solution was removed by centrifuging tubes at 11,363xg for 20 min at 4°C in a microcentrifuge; the lower fraction was removed with an 18-gauge needle from below the floating lipid droplet fraction. Lipid droplet fractions in microcentrifuge tubes were delipidated with 2 l of cold acetone overnight at −80°C, followed by centrifugation at 11,363xg for 30 min at 4°C, and removal of solvent from the protein pellet. The pellet was further extracted with acetone at room temperature, followed by 1:1 acetone:ether (v:v), and finally ether.

Residual solvents were allowed to evaporate completely and lipid droplet fractions were stored at −80°C. Mass spectrometry was performed on four separate sample sets, each set consisting of lipid droplets isolated from iBAT pooled from 2 to 3 age-paired *Alk7*⁻/⁻ and wild type mice. Delipidated lipid droplet pellets were resuspended using urea, sonication and vortexing. A tryptic digestion of 10 µg was carried out with Urea/proteaseMax protocol for subsequent nLC-MS/MS analysis on QExactive, long gradient and nLC II. A standard proteome quantitation analysis was then performed.

Primary culture of mouse and human brown adipocytes iBAT from 4 to 6 month old mice was dissected out, micd and digested for 60 min in DMEM/F12 medium supplemented with 1% BSA, antibiotics, and 1 mg/ml collagenase II (Sigma Aldrich) under constant shaking. The digested tissue was filtered through 250 µm nylon mesh and a 70 µM cell strainer and centrifuged for 10 min at 1500 rpm to separate floating mature adipocytes. The pellet was resuspended in erythrocyte lysis buffer (15 mM NH4Cl, 10 mM KHCO3, 0.1 mM EDTA) for 10 min to remove blood cells. The cells were further centrifuged at 1500 rpm for 10 min, and the pellet (stromal vascular fraction, SVF) was resuspended and plated in 48-well plates. Cells were grown in DMEM/F12 supplemented with 10% FBS and 100 µg/ml penicillin-streptomycin at 37°C until confluence. Adipocyte differentiation was induced (day 0) in medium supplemented with 1 µM dexamethasone, 66 nM insulin, 15 mM HEPES, 1nM T3, 33 µM biotin, 17 µM pantothenate, 10 µg/ml transferrin, and 1 µg/ml rosiglitazone until full differentiation (day 10).

Human brown preadipocytes (kindly provided by Dr. Shingo Kajimura) were differentiated as previously described (*Shinoda et al., 2015*). Cells were grown in maintenance medium (Advanced DMEM/F12 supplemented with 10% FBS, 100 µg/ml penicillin-streptomycin, 1nM T3, 2 µg/ml dexamethasone and 5 µg/ml insulin) at 37°C until confluence. Differentiation was started (day 0) with induction medium (Advanced DMEM/F12 supplemented with 10% FBS, 100 µg/ml penicillin-streptomycin, 1nM T3, 2 µg/ml dexamethasone, 5 µg/ml insulin, 1 µM rosiglitazone, 0.125 mM indomethacin, and 0.25 mM IBMX) in collagen-coated 48-well plates until full differentiation (day 24).

For activin B treatment, fully differentiated mouse or human brown adipocytes were incubated with 100 ng/ml activin B (R and D Systems) in DMEM/F12 supplemented with 10% FBS and 100 µg/ml penicillin-streptomycin for 24 hr before lysis and for RNA extraction of RNA (as indicated below). SB-431542 (Sigma-Aldrich) was added at the same time at a final concentration of 10 µM, as indicated.

## RNA isolation, real-time quantitative PCR and microarray analysis

RNA from tissue and cells was extracted using the RNeasy Mini Kit (Qiagen), treated with DNase I (Life Technologies), and reversely transcribed using SuperScript II reverse transcriptase (Life Technologies). cDNAs were used for real-time quantitative PCR (Q-PCR) using the primers listed in *Supplementary file 1*. Q-PCR was performed with SYBR Green PCR Master Mix (Life technologies) on a StepOnePlus real time PCR system (Applied Biosystems), using 18S rRNA as an endogenous normalization control. For quantification of mitochondria DNA, total tissue DNA was extracted with a DNeasy kit and used for real-time quantitative PCR using primers for the mitochondrial gene *CytB*, encoding cytochrome B (*Guo et al., 2014*). Microarray analysis was performed on mRNA extracted from iBAT of 6 month old wild type and *Alk7*⁻/⁻ mice that had been kept 4 months at 30°C with ad libitum Chow feeding, using microarray chips (MouseRef-8_v2_0_R3_11278551_A) from Illumina and following the manufacturer's instructions.

## Western blotting

Snap-frozen tissues were homogenized with a high-speed tissue homogenizer in ice-cold RIPA buffer (50 mM Tris-HCl, 150 mM NaCl, 5 mM EGTA, 1% NP-40, pH7.4) supplemented with proteinase and phosphatase inhibitor cocktails (Roche), and centrifuged at 845xg for 15 min at 4°C to collect supernatants. Protein concentration was determined by the BCA method. Supernatants were used for reducing SDS-PAGE, transferred onto PVDF membranes (Amersham) and analyzed by western blotting using the specific first antibodies listed in *Supplementary file 2* with goat anti-rabbit and goat anti-mouse peroxidase-conjugated immunoglobulin as secondary antibodies (Dako). The blots were then processed with the luminescence technique ECL (Thermo Scientific), and imaged in an Imagequant LAS4000. The levels of target proteins were quantified by the intensity of western blot bands using ImageJ software (National Institutes of Health), using actin as loading control.

## Statistical analyses

Statistics analyses were performed using Prism five software (GraphPad, SPSS IBM corporation) and Microsoft Excel (Microsoft). Student's t test, one-way ANOVA or two-way ANOVA were performed to test statistical significance according the requirements of the experiment. In some cases, two-tailed Mann Whitney was used as a non-parametric test. Bonferroni or Tukey's post-tests were used, when indicated, as a further test for experiments that required multiple comparisons. The level of statistical significance was set at $p < 0.05$ for all the analyses (*). All P values are reported in the figure legends.

## Acknowledgements

We thank Annika Andersson (Karolinska Institute) for help with mice genotyping; Dorothea Rutis-hauser (Karolinska Institute) for help with mass spectrometry analysis of lipid droplet proteomes; Julie Massart and Marie Björnholm (Karolinska Institute) for help with metabolic cages; Boon Seng Soh, Kian Leong Lee, Henry Yang He, and Bing Lim (ASTAR, Singapore) for help with microarray studies; Jan Nedergaard and Barbara Cannon (Stockholm University) for advice and help with preliminary studies; Shingo Kajimura for human brown preadipocytes; and Evan Rosen (Harvard Medical School) for $Ucp1^{CRE}$ mice. Support for this research was provided by grants to CFI from the Swedish Research Council, Swedish Cancer Society, Knut and Alice Wallenbergs Foundation (Wallenberg Scholars Program), National Medical Research Council of Singapore and National University of Singapore.

## Additional information

### Funding

| Funder | Grant reference number | Author |
| --- | --- | --- |
| Vetenskapsrådet | 2016-01538 | Carlos F Ibáñez |
| Cancerfonden | 180670 | Carlos F Ibáñez |
| National University of Singapore | R-185-000-282-720 | Carlos F Ibáñez |
| National Medical Research Council | NMRC/CBRG15may004 | Carlos F Ibáñez |
| Swedish Research Council | | Carlos F Ibáñez |
| Swedish Cancer Society | | Carlos F Ibáñez |
| Knut och Alice Wallenbergs Stiftelse | Wallenberg Scholars Program | Carlos F Ibáñez |

The funders had no role in study design, data collection and interpretation, or the decision to submit the work for publication.

### Author contributions

Patricia Marmol, Conceptualization, Data curation, Supervision, Funding acquisition, Investigation, Methodology, Writing - original draft, Project administration, Writing - review and editing; Favio Kra-pacher, Formal analysis, Investigation, Methodology, Writing - original draft; Carlos F Ibáñez, Conceptualization, Data curation, Formal analysis, Supervision, Funding acquisition, Investigation, Project administration, Writing - review and editing

### Author ORCIDs

Carlos F Ibáñez (iD) https://orcid.org/0000-0003-4090-0794

### Ethics

Animal experimentation: Animal protocols were approved by Stockholms Norra Djurförsöksetiska Nämnd (Stockholm North Ethical Committee for Animal Research) and are in accordance with the ethical guidelines of the Karolinska Institute. Protocol numbers 111/14, 8935-17 and 36/14.

### Decision letter and Author response

Decision letter https://doi.org/10.7554/eLife.54721.sa1
Author response https://doi.org/10.7554/eLife.54721.sa2

## Additional files

### Supplementary files

• Supplementary file 1. PCR primers. The PCR primers used in this study are listed here.

• Supplementary file 2. Antibodies. The primary antibodies used in this study are listed here.

• Transparent reporting form

### Data availability

All data generated or analysed during this study are included in the manuscript and supporting files.

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
