## [Decision Letter]

**Acceptance summary:**

The authors show that *Alk7* inactivation in interscapular brown adipose tissue (BAT) resulted in negligible phenotypic differences when mice were reared in standard dietary and environmental conditions. However, fasting of mice resulted in reduced size of lipid droplets, and increased Atgl expression and lipolysis in BAT of the BAT-specific *Alk7* deficient mice. These mice also showed an increased expression of genes associated with nutrient stress (e.g. Klf15, Prodh) which was shown to be mitigated if mice were treated with the glucocorticoid receptor (GR) antagonist RU486. Thus, data suggests that *Alk7* inactivation in BAT leads to excessive glucocorticoid signaling upon fasting. Overall, results show that ALK7 plays an integral role in how BAT adapts to nutrient availability which may aid in the development of novel strategies to utilize BAT energy expenditure for the treatment of metabolic disorders such as obesity and T2D.

**Decision letter after peer review:**

Thank you for submitting your article "Control of brown adipose tissue adaptation to nutrient stress by the activin receptor ALK7" for consideration by *eLife*. Your article has been reviewed by three peer reviewers, including Rob Koza as the Reviewing Editor and Reviewer #1, and the evaluation has been overseen by Clifford Rosen as the Senior Editor. The following individual involved in review of your submission has agreed to reveal their identity: Minho Shong (Reviewer #3).

The reviewers have discussed the reviews with one another and the Reviewing Editor has drafted this decision to help you prepare a revised submission.

Summary:

A previous study (Guo et al., 2014) by the corresponding author showed that global or adipose specific inactivation of TGB-beta superfamily receptor ALK7 reduces high fat diet induced catecholamine resistance and the development of obesity in mice. These actions occur presumably through enhanced energy expenditure due to increased beta-adrenergic signaling. This current study is an extension of the previous study in that the authors used a similar approach to inactivate *Alk7* specifically in interscapular brown adipose tissue (BAT) of mice, which resulted in negligible phenotypic differences in mice reared in standard vivarium conditions. However, fasting of mice for 14 hours resulted in reduced size of lipid droplets and increased expression of Atgl and lipolysis in BAT of *Alk7* deficient mice. Mice with BAT deficient in *Alk7* also showed increased expression of genes associated with nutrient stress (e.g. Klf15, Prodh). Treatment of mice with BAT-specific inactivation of *Alk7* with the glucocorticoid receptor (GR) antagonist RU486 resulted in reduced expression of Klf15 and PRODH and suggests that *Alk7* inactivation in BAT causes excessive glucocorticoid signaling upon fasting.

Overall, results showing that ALK7 plays an integral role in how BAT adapts to nutrient availability may aid in the development of novel approaches to utilize BAT energy expenditure for the treatment of metabolic disorders such as obesity and T2D.

Although the studies by the authors are thorough and convincing, and the manuscript presented clearly and logically, the reviewers have agreed that the authors address several concerns prior to acceptance.

Essential Revisions:

1) It is requested that schematic model for the proposed signaling pathway related to the role of ALK7 in mediating nutrient stress in BAT be included.

2) A discussion of potential mechanisms for upregulation of the hypothalamus-pituitary axis in *Alk7*-deficient mice.

3) There are many caveats with respect to the potential lack of specificity of RU486 because of its effect on a plethora of plasma molecules. Because of this concern, the reviewers recommend that the authors include a concise and thoughtful discussion of these caveats with respect to the experimental model.

4) Although evidence is suggestive of a role for KLF15 in the regulation of POX and ATGL, it is not conclusive. The reviewers recommend that the authors either thoroughly discuss, or preferentially provide more evidence of a direct role for KLF15 regulation of POX and ATGL.

5) There are some concerns that mRNA expression for several of the genes (e.g. Prodh, Alt1, Bcat2 and Klf15) may not accurately reflect protein expression. The reviewers recommend the measurement of protein levels for these genes if good antibodies are readily available. If good antibodies are not available, at least address the potential for discordancy between gene transcription and translation.

6) The reviewers suggest that *Adrb1* and *Adrb3* mRNA are measured in conditions where corticosterone is high (Figure 3D) to completely rule out a similar mechanism previously published by the authors (Guo et al., 2014).

---

## [Author Response]

Essential Revisions:1) It is requested that schematic model for the proposed signaling pathway related to the role of ALK7 in mediating nutrient stress in BAT be included.

Schematic is presented in Figure 6—figure supplement 5.

2) A discussion of potential mechanisms for upregulation of the hypothalamus-pituitary axis in Alk7-deficient mice.

The mutant mice used in the present study lack *Alk7* only in brown adipocytes. The Ucp1-CRE is very specific to brown adipocytes and will not delete *Alk7* in the hypothalamus or pituitary. As shown in Figure 3F, there is no change between genotypes in the circulating levels of corticosterone, indicating that the effects of *Alk7* deletion in brown adipocytes are local and likely due to change in sensitivity to glucocorticoids. We would not expect any changes in hypothalamus-pituitary axis given that glucocorticoid levels appeared not to be changed in our mutant mice.

3) There are many caveats with respect to the potential lack of specificity of RU486 because of its effect on a plethora of plasma molecules. Because of this concern, the reviewers recommend that the authors include a concise and thoughtful discussion of these caveats with respect to the experimental model.

As requested, we have included a discussion of this problem in the Results section, as follows: “We note that, in addition to function as a GR antagonist, RU486 has been shown to exert complex effects on a range of other steroid receptors, most notably as an antagonist of the progesterone receptor (Lin et al., 2001 and Zhang et al., 2006). However, levels of progesterone in serum follow a circadian pattern, and reach a peak during night time in male rodents (such as the mice used here) (Kalra et al., 1977). By administering RU486 in the morning, we believe that such effects were minimized in our studies.”

4) Although evidence is suggestive of a role for KLF15 in the regulation of POX and ATGL, it is not conclusive. The reviewers recommend that the authors either thoroughly discuss, or preferentially provide more evidence of a direct role for KLF15 regulation of POX and ATGL.

There is ample evidence of the role of KL15 in regulating POX and ATGL in the literature. Those links have now been included in the schematic (Figure 6—figure supplement 5) and in the Discussion as requested.

5) There are some concerns that mRNA expression for several of the genes (e.g. Prodh, Alt1, Bcat2 and Klf15) may not accurately reflect protein expression. The reviewers recommend the measurement of protein levels for these genes if good antibodies are readily available. If good antibodies are not available, at least address the potential for discordancy between gene transcription and translation.

The protein levels of POX, encoded by the Prodh gene, were already reported in our original manuscript (original Figures 5A and B). The revised version now includes protein data for KLF15, ALT1 and BCAT2 (revised Figures 3D and E). Commercial antibodies to KLF15 are not very good and were difficult to work with. However, after three experiments each in triplicate, results are very consistent. In all cases, fasting induces a larger increase in mice lacking ALK7 in brown adipose tissue compared to control mice, as expected from the mRNA data. We have also included data showing upregulation of C/EBPα (revised Figure 2F), a well known regulator of adipocyte differentiation which has been shown to collaborate with PPARγ to upregulate ATGL expression in adipocytes, establishing another possible route for the effects of ALK7 on ATGL expression.

6) The reviewers suggest that Adrb1 and Adrb3 mRNA are measured in conditions where corticosterone is high (Figure 3D) to completely rule out a similar mechanism previously published by the authors (Guo et al., 2014).

We did not detect any significant change between genotypes in *Adrb1* or *Adrb3* mRNA levels upon fasting (high corticosterone condition) and this is now shown in Figure 2—figure supplement 1C and D.